# Matrix Metalloproteinases and Glaucoma

**DOI:** 10.3390/biom12101368

**Published:** 2022-09-25

**Authors:** Moo Hyun Kim, Su-Ho Lim

**Affiliations:** 1Department of Ophthalmology, Daegu Premier Eye Center, Suseong-ro 197, Suseong-Gu, Daegu 42153, Korea; 2Department of Ophthalmology, Daegu Veterans Health Service Medical Center, 60 Wolgok-Ro, Dalseo-Gu, Daegu 42835, Korea

**Keywords:** glaucoma, matrix metalloproteinases, trabecular meshwork, ocular surface, wound healing, genetics

## Abstract

Matrix metalloproteinases (MMPs) are enzymes that decompose extracellular matrix (ECM) proteins. MMPs are thought to play important roles in cellular processes, such as cell proliferation, differentiation, angiogenesis, migration, apoptosis, and host defense. MMPs are distributed in almost all intraocular tissues and are involved in physiological and pathological mechanisms of the eye. MMPs are also associated with glaucoma, a progressive neurodegenerative disease of the eyes. MMP activity affects intraocular pressure control and apoptosis of retinal ganglion cells, which are the pathological mechanisms of glaucoma. It also affects the risk of glaucoma development based on genetic pleomorphism. In addition, MMPs may affect the treatment outcomes of glaucoma, including the success rate of surgical treatment and side effects on the ocular surface due to glaucoma medications. This review discusses the various relationships between MMP and glaucoma.

## 1. Introduction

Matrix metalloproteinases (MMPs) are proteolytic enzymes characterized by zinc ions at the catalytic site and cysteine switches in the propeptide region. They are secreted as a latent proenzyme that must be activated before it functions via other intra- or extracellular enzymes [1,2]. After activation, MMPs cleave a wide range of extracellular matrix (ECM) structures (collagen, gelatin, proteoglycans, laminin, and fibronectin) in the extracellular environment. In addition to ECM structural remodeling, MMPs are also involved in the regulation of cellular functions. They regulate several cell surface receptors and growth factors, chemokines, cytokines, and cell-to-cell adhesion molecules [3,4,5,6]. Due to these functions, MMPs are involved in the overall environmental composition of the ECM and thus influence extracellular activities such as apoptosis, cell proliferation, and cell migration [2,3,7,8,9]. MMP activity is antagonized by TIMPs, α2-macroglobulin, and reversion-inducing cysteine-rich proteins with Kazal motifs. Four groups of TIMPs have been identified that bind MMPs with variable affinities at a 1:1 stoichiometric ratio. In addition to these regulatory factors, post-translational control, cell compartmentalization, and other mechanisms modulate MMP activity [10,11]. In addition to ECM remodeling, which is the main function of MMPs, MMPs are expressed in various other tissues and cells, suggesting that MMPs affect homeostasis in a wide range of cells and organs. Therefore, when MMP activity is properly controlled, MMPs play an important role in tissue remodeling processes, such as angiogenesis, neural plasticity, organogenesis, and wound healing. However, unregulated MMP activity, such as unbalanced MMP and TIMP activities, can cause many pathological conditions. Metastatic activity in cancer, cardiovascular disorders, neurodegenerative disorders, and many other diseases are related to impaired MMP activity [12,13,14,15,16,17] (Figure 1).

MMPs can be subdivided according to their sequence similarity, domain organization, and proteolytic activity. Collagenase groups (MMP-1, -8, and -13) can specifically cleave collagen with gelatinases in a synergistic manner. The gelatinase group (MMP-2 and -9) plays many roles by degrading collagens, elastin, and aggrecan and regulating cell signal activity by controlling cytokines and chemokines. The stromelysin group (MMP-3, -10, and -11) is involved in proteoglycan, laminin, and fibronectin degradation but cannot cleave type I collagen. The matrilysin group (MMP-7 and -26) is involved in fibronectin, elastin, and actin degradation and regulates the activity of many cytokines. The membrane-type group (MMP-14, -15, -16, -17, -24, and -25) activates other MMPs and exhibits proteolytic activity against several ECM components. Finally, other groups of MMPs are cell- or tissue-specific enzymes that are not routinely synthesized under specific conditions [18,19] (Table 1).

Glaucoma is a neurodegenerative disorder affected by multiple factors, from which more than 60 million patients worldwide suffer [30]. It is characterized by irreversible progressive loss of retinal ganglion cells (RGC) and distinct optic nerve head (ONH) deterioration, which is related to corresponding visual field loss [31,32,33]. The prevalence of glaucoma increases with age, and elevated intraocular pressure (IOP) is the most important factor in its development and progression. Therefore, lowering the IOP is the most important clinical treatment. However, there are factors that affect glaucoma that cannot be explained by IOP alone, because glaucoma progresses even when intraocular pressure is well controlled [34,35]. Despite the efforts of many researchers, the exact mechanism by which elevated IOP and multiple other factors affect glaucoma and cause RGC apoptosis and progression at the molecular or cellular levels is little understood. Although some animal clinical studies have shown results that can partially explain the mechanisms underlying the pathogenesis of glaucoma, they often show conflicting results owing to various limitations [36,37,38,39].

Despite the unknown effects of various environmental factors, MMPs are important because of their possible association with the etiology of glaucoma, glaucoma type, glaucoma surgery outcomes, and other eye diseases. In this context, we reviewed the association between MMPs and glaucoma [30,40,41,42,43,44].

This review aimed to investigate the overall relationship between MMPs and glaucoma, focusing on genetic pleomorphism, pathogenesis, wound healing during surgical treatment, and ocular surface changes during medical treatment.

PubMed and Google Scholar were the databases used for literature research, and the following terms were queried: “glaucoma,” “metalloproteinases,” “MMP,” and “glaucoma AND metalloproteinases”. Systematic reviews, narrative reviews, meta-analyses, and clinical trials published over the past 15 years were also included. Titles and abstracts of the searched articles were screened to identify potentially eligible studies. After screening, all relevant studies were assessed in detail by examining their full texts. The bibliographies of all the related articles were examined to identify additional related studies. Studies were excluded if they were written in languages other than English and if they were not related to the research question. All the processes presented above were performed by two authors (MH Kim and SH Lim), as shown in Figure 2.

Initially, 328 studies published within the last 15 years were retrieved from PubMed and Google Scholar databases. Titles, abstracts, and relevant articles from the literature were reviewed. In total, 160 eligible full-text articles were identified. The current results regarding genetic pleomorphisms in MMPs and glaucoma, glaucoma pathogenesis concerning MMPs, MMPs, and wound healing in glaucoma surgical treatments, and ocular surface changes due to MMP function during glaucoma medical treatment are described below.

## 2. Results and Discussion

### 2.1. Pathogenesis of Glaucoma concerning Matrix Metalloproteinases

#### 2.1.1. Pathogenesis of Glaucoma Subtypes

Primary angle-closure glaucoma (PACG) is caused by compromised aqueous humor (AH) outflow and anterior chamber angle (ACA) obstruction. In most cases, the PACG angle narrows because of the relative pupillary block, which is related to displacement of the peripheral iris against the trabecular meshwork (TM) [45]. PACG prevalence differs according to ethnicity, sex, and family history. Studies have reported that PACG occurs three times more frequently in Asian populations than in European populations [46,47], and females are more prone to develop PACG [48,49]. Studies involving Chinese and Eskimos have shown that individuals with any first-degree relative with PACG are more susceptible to developing PACG [50,51].

Pseudoexfoliation glaucoma (XFG), which is caused by pseudoexfoliation syndrome (XFS), is the most commonly identifiable cause of open-angle glaucoma and is caused by pseudoexfoliation syndrome (XFS) [52]. XFS is an age-related syndrome characterized by the deposition of white scale-like substances in ocular tissues [53]. Scale-like substances are called exfoliation materials (XFM) and produce an abnormal accumulation of fibrillary elastic ECM [54]. The pathological events of XFG have not been fully confirmed; however, it is assumed that IOP is increased by AH outflow obstruction, which is induced by the deposition of XFM in the TM structure, finally leading to glaucomatous changes in the ONH [54]. Studies involving twins and first-degree relatives with XFS and loss of heterozygosity have indicated that XFG and XFS exhibit strong familial inheritance [55,56,57,58].

Juvenile open-angle glaucoma (JOAG) is an uncommon type of primary open-angle glaucoma (POAG), with a high prevalence in individuals aged 5–35 years. Individuals with JOAG follow autosomal dominant inheritance and have a strong family history of POAG [59]. Myocilin gene (MYOC) mutations account for approximately 10% of JOAG cases [31,60]. It has been suggested that MYOC mutations disrupt MMP and TIMP activities in the TM, causing pathological changes in the development of glaucoma [61,62].

#### 2.1.2. Matrix Metalloproteinases and Trabecular Meshwork with Glaucoma

Elevated IOP is one of the main risk factors for glaucoma [34,35]. IOP is defined as the difference between the production of aqueous humor (AH) in the ciliary body and AH drainage, mainly in trabecular meshwork (TM) and minorly in the uveoscleral pathway [63,64] (Figure 3). The TM generates the main outflow resistance for the AH, which is positioned at the iridocorneal angle, and its ECM is constantly remodeled by MMPs. ECM composition is continuously remodeled by selective ECM substrate degradation and production of new ECM substrates, including fibronectin, proteoglycans, collagens, and glycosaminoglycans, by TM cells. MMPs play a role in degrading specific ECM substrates [63,65]. Along with ECM modification, the geometry of the TM changes to regulate permeability via ciliary muscle contraction and form changes in the TM cells [66,67,68]. Many MMPs (MMP-1, -2, -3, -9, -12, and -14) and their local inhibitor TIMP-2 have been synthesized by TM cells [69,70]. Increasing MMP activity causes an elevated AH outflow rate, whereas inhibiting MMP activity decreases the AH outflow rate [71]. A recent study using a porcine model showed similar results, indicating that reduced MMP-2 and -9 activity is related to elevated IOP [72]. ECM composition changes can be observed when the AH outflow rate is pushed from equilibrium [73].

When the outflow of AH decreases, IOP increases, which causes mechanical stretching of the TM, owing to pressure changes. This structural change is sensed by TM cells, which trigger homeostatic activity to reduce the IOP. TM cells increased the synthesis of MMPs (MMP-2, -3, and -14) and decreased TIMP-2 production. Altered MMP and TIMP activity causes an increased ECM turnover rate with degraded ECM substrate uptake, changing ECM biosynthesis and changing the ECM environment to adjust outflow resistance. After ECM remodeling, the AH outflow rate increased and returned to normal IOP. In in vitro studies, adding recombinant MMPs to human TM organ cultures induced an elevated AH outflow rate; however, inhibiting MMP activity decreased the outflow rate [66,71,74,75,76,77,78]. This suggests that mechanical changes in the TM function as sensors of IOP change. ECM remodeling does not occur by simply releasing endogenous MMPs into the extracellular regions. Aga et al. [79] discovered the detailed regulatory areas. This region, called the podosome- or invadopodia-like structure (PILS), is localized in distinct TM cell areas. Increased MMP-2 and -14 expression has been observed in PILS, and these structures function in cell attachment and ECM turnover in a highly controlled manner. PILSs play an important role because uncontrolled ECM remodeling in the main AH outflow route may disrupt ECM structures, causing dysregulation of outflow resistance. Specifically, in an in vitro study on animal cell cultures, this change in MMP protein levels was not related to changes in mRNA transcription. This suggests that TM cells sense mechanical stretching through interactions between integrin and ECM, which induces signal transduction through signal transduction inhibitors (rapamycin and wortmannin). Finally, ribosomes are recruited to the mRNA to translate MMP-2 and -14 [40,74].

Disruption of MMP activity has been reported in glaucoma patients. Microscopic TM findings in patients with primary open-angle glaucoma (POAG) show a significant depletion of hyaluronic acid (HA) compared with those without the disease. Normally, MMP-2 and -9 mRNA activity increases HA concentration. Therefore, in patients with POAG, it can be assumed that HA depletion is a result of reduced MMP activity and causes disruption of ECM remodeling, leading to decreased AH outflow rates [80,81]. Unlike the decreased activity of MMPs in TM, other studies have suggested that increased MMP and TIMP activities are observed in the AH of patients with POAG [82,83].

Corticosteroids are currently used to treat various ophthalmic diseases [84]. In most cases, when used with caution, there are few side effects, but some patients develop steroid-induced glaucoma [85]. In an in vitro study of human TM cells, dexamethasone decreased MMP-2 and -9 activity compared with the non-dexamethasone control [86]. Another study using human fibroblasts reported similar results [87]. Reduced expression of MMP-2 and -9, which are caused by corticosteroids, may decrease AH outflow and increase IOP, resulting in steroid-induced glaucoma.

#### 2.1.3. Matrix Metalloproteinases and Neuroretina with Glaucoma

MMPs play a major role in regulating AH outflow resistance by remodeling the ECM in the TM. However, MMPs are also factors that contribute to degenerative conditions in the posterior segment of the eye in glaucoma. Many MMPs and their inhibitors, TIMPs, are expressed in various neuronal and glial cells of the retina and the optic nerve. This expression may affect glaucomatous retinal and optic nerve degeneration by inducing apoptosis of retinal ganglion cells (RGC) and atrophic changes in the optic nerve [88,89,90,91,92,93]. However, the exact mechanism by which elevated IOP induces retinal degeneration and optic nerve atrophy remains unclear.

Approximately 2 million RGC axons merge at a point in the posterior segment of the eye known as the optic nerve head (ONH). RGC axons at the ONH enter the neural canal and penetrate the Bruch’s membrane, choroid, and sclera [94]. As the RGC axon bundle passes through the ONH, a mesh-like structure called the lamina cribrosa (LC) supports the RGC axon bundles as they become the optic nerve [95,96]. In addition to the structural support of RGC axons, the LC has capillaries and glial cells within its structure that manage the ECM environment and provide nutritional support to RGC axons. As RGC axons forming the optic nerve are primarily supported by the LC, they are the main site within the rigid corneoscleral envelope, which is affected by the mechanical stress induced by increased IOP. Mechanical stress in the LC activates connective tissue remodeling cascades involving glial cells, LC cells, and scleral fibroblasts. Glial cells in the LC increase MMP secretion and modulate ECM structures to resemble glaucomatous environments. When ECM remodeling is complete, compression of RGC axons is relieved, and axoplasmic flow continues, which is affected by mechanical stress in the LC [97,98]. LC and glial cells are very sensitive to changes in mechanical stress that affect LC because they can sense mechanical stress by integrin receptors that are linked directly to the fibrillar ECM with cytoskeletons [99].

Many studies have investigated MMP and TIMP activities and expression in various glaucomatous animal models and humans with glaucomatous optic nerve changes. These studies suggest a mechanism by which MMPs affect glaucomatous optic nerve degeneration. In glaucomatous optic nerves, increased IOP induces mechanical stress in the LC, which is sensed by LC and glial cells. These cells secrete more cytokines (TGF-b1 and TNF-α) than normal cells do. Increased cytokine activity induces MMP-2 expression and ECM remodeling in the ONH. In addition, glial cells transform into reactive rather than quiescent forms and express MT1-MMP and MMP-1. MMP-1 causes continuous ECM degradation if it is not inhibited by TIMP-1, which is secreted by glial cells and RGC. If degradation continues, the LC cannot support RGC axon survival and may induce ONH excavation [41,91,92,97,98,100]. In patients with POAG and normal tension glaucoma (NTG), MMPs (MMP-1, -2, and -3) were enriched in the ONH [39,91,92] (Figure 4).

Aging is another well-known risk factor for the development and progression of glaucoma in addition to elevated IOP. The composition of LC ECM proteins and collagen changes with age. Owing to these changes, the laminar beam becomes thicker with increased collagen and elastin within the cribriform plates [101,102]. Thickened laminar beams decrease LC elasticity and impede the diffusion of nutrients to the RGC axons. It is more common for aged ONH patients to suffer damage from elevated IOP or other non-IOP related situations [103,104,105]. Increased collagen and elastin levels have been suggested to be responsible for the decreased MMP activity. However, a study with aged human donor eyes showed that all members of the gelatinase family (MMP-2, -9), which are found in the ECM of Bruch’s membrane, are also observed in the ONH [44]. Their results showed that the level of active MMPs was higher than that of the proenzymes. This indicates that the degradation process leads to the regeneration of the LC under normal conditions with or without aging. However, Hussain et al. have suggested that age-related collagens are insoluble, chemically altered, and difficult to degrade. In addition, aged LCs have more advanced glycation end-products, which are powerful inhibitors of MMPs, making enzymatic breakdown more difficult.

MMPs play an important role in the regulation of the ECM of the retina, similar to many other human tissues. The interaction between ECM and MMPs affects RGC survival [39]. Specifically, elevated MMP-9 levels expressed by RGCs or reactive astrocytes induce RGC apoptosis by inducing detachment and promoting laminin proteolysis [39,106]. Animal models of RGC death show elevated MMP-9 activity, and most MMP-9 knockdown mouse models with optic nerve ligation do not show RGC death [90,107,108,109,110,111,112]. However, the role of MMP-2 in animal models of RGC death remains controversial. A few studies [89,90,109,110] have reported unchanged MMP-2 activity after RGC death, but others have found that MMP-2 activity increased after excitotoxic injury or post-ischemia-reperfusion [108,113].

The precise mechanism of MMP and TIMP activities in glaucomatous retinas remains elusive and requires further detailed studies. However, some studies using animal and human cell culture models of glaucoma have provided insights into its activity. These studies suggest that events related to glaucomatous changes in the retina (ischemic insults and increased IOP) cause changes in the cellular signal transduction. Elevated cytokine (IL-1) and retinal glutamate levels stimulate MMP-9 synthesis in RGCs and glial cells. IL-1 also stimulates nitric oxide production, which activates dormant extracellular pro-MMP-9. In the ECM, TGF-β2, a well-known MMP inhibitor, is decreased by an increase in the IOP. Through this process, MMP-9 activity increases in the RGC layer and induces apoptosis of RGCs via many cellular signaling cascades. In contrast, in an ocular hypertension model, TIMP-1 showed increased activity, followed by elevated MMP-9 expression, and inhibited the activity of MMP-9 to prevent apoptosis of RGCs. In addition, RGC apoptosis may be stopped by TIMP-1 activity by inhibiting signals that induce RGC apoptosis in a non-MMP-related manner [39,41,108,109,110,114,115].

### 2.2. Genetic Polymorphism in Matrix Metalloproteinases and Glaucoma Subtypes

Genetics have also been suggested to be an important risk factor for glaucoma. Individuals with a family history of glaucoma have an increased risk of developing glaucoma [116,117]. Some studies on genetics at the molecular level have shown that genetic alterations such as single nucleotide polymorphisms (SNPs) can change gene expression activity, and the function of genetics can alter the level of gene expression or the function of genetic compounds. Alterations in SNPs may change susceptibility to diseases in individuals [118,119]. Few studies have identified genetic loci that may be related to glaucoma risk, but these findings do not explain more than 10% of glaucoma heritability [117]. 

Several studies of the relationship between MMP family genetic polymorphisms and glaucoma risk have been conducted in many ethnicities. Most studies have focused on MMP-9, and only a few studies have focused on MMP-1 and -2. When viewed in relation to glaucoma subtypes and SNP mutations, most were limited to POAG and PACG. There are few studies on the remaining subtypes; therefore, the interpretation of the results is very limited. Due to these limitations, several studies have reported conflicting results.

The rs1799750 is SNP at position -1607 of the promoter of the MMP1 gene, which has an insertion polymorphism. The variants can have “2G” insertion polymorphism, which leads to elevated levels of MMP-1 in the serum, potentially to facilitate collagen breakdown rather than the 1G genotype [120]. Many studies involving two meta-analyses have reported an increased risk of developing POAG in Asians and Caucasians with rs1799750 [121,122,123,124,125]. One meta-analysis suggested that rs1799750 is associated with a risk of developing PACG and XFG [123]. In addition to the increased risk of POAG, Ponomarenko et al. [126] reported that rs1799750 is related to earlier onset of POAG development. However, Mossböck et al. [127] reported that G was not associated with POAG and XFG development in Caucasians. Another study involving a Greek population showed that rs1799750 was only related to pseudoexfoliation syndrome without glaucomatous changes [128]. The clinical characteristics of rs1799750 in glaucoma are likely to cause optic nerve head damage [91] and are likely to increase AH outflow resistance by abnormal accumulation of ECM in the TM of patients with POAG [125]. Essential hypertension is a risk factor for glaucoma [129] and rs1799750 increases the risk of developing hypertension [130].

Altered MMP-9 gene expression may affect AH outflow imbalance in the TM and glaucomatous changes in the optic nerve head [71,106]. 

Several studies on rs3918249 have reported a decreased risk of glaucoma [126,131,132]. A reduced risk of POAG has been reported in the Russian population [126]. Two studies reported a decreased risk of XFG and glaucoma in Caucasians [131,132]. However, two other studies reported that PACG risk was increased by rs3918249 [133,134], and the increased risk of PACG by rs3918249 likely to decrease the axial length of the eye in PACG patients [134]. However, two studies in a Chinese population showed that rs3918249 is not related to PACG [135,136].

Few studies regarding rs2250889 have shown an increased risk of developing glaucoma, and none have shown a protective effect [126,132,137]. In the Russian population, rs2250889 increased the risk of POAG, but interestingly, IOP was lower in the POAG population that had a relationship with rs2250889 [126]. Another study in Russia reported that rs2250889 increased the risk of XFG in Caucasians [132]. In Chinese individuals, PACG incidence was higher with rs2250889, and elevated MMP-9 plasma levels were noted, but the incidence of POAG was not related to rs2250889 [137].

Many studies have been published on rs17576. As the number of studies was large, the results were conflicting. The risk of PACG was high in Pakistan [122], China [137], and Australia [133], with rs17576. Furthermore, rs17576 increases the risk of developing POAG only in the Chinese population [137]. However, other studies have reported that rs17576 has no relationship with POAG and PACG [127,131,136,138]. In Russian individuals [126], rs17576 decreased the risk of POAG and reduced IOP in POAG patients with rs17576. Caucasians with rs17576 showed a decreased risk of developing glaucoma [139]. In addition to the direct risk of glaucoma development, rs17576 is likely to increase hypertension [140] and diabetes incidence [141], which are other factors that increase the risk of glaucoma development [129,142].

Gao et al. [135] reported that rs3918254 increased the risk of developing PACG in Chinese individuals, but three other studies, including meta-analyses, showed that rs3918254 was not related to PACG risk [133,139,143].

For rs2274755, only one study on glaucoma has been published [144]. Suh et al. [144] reported that rs2274755 is significantly associated with NTG in Koreans. Additionally, rs2274755 is likely to cause weakness in the lamina cribrosa of the optic nerve head, which is associated with glaucomatous optic nerve head changes.

Caucasians with POAG and Indian patients with POAG or PACG have a significant relationship with rs3918242 [124,145]. However, in a Russian population, rs3918242 decreased the risk of developing POAG [126]. A meta-analysis by Chen et al. [125] showed that POAG was not associated with rs3918242, and two other studies, including the Chinese [137] and Iraqi [146] studies, reported similar results. Although Zhao et al. [137] showed that rs3918242 was not associated with POAG and PACG, rs3918242 was significantly associated with elevated MMP-9 levels in the plasma (Table 2).

Single SNPs may affect phenotypes but are not responsible for the genetic sensitivity of the disease [147]. Therefore, SNP interactions can work synergistically to become stronger risk factors [147]. One study reported that the disease did not manifest itself because of each SNP, but appeared when it worked together [148]. In addition, the degree of disease expression is different not only in the SNP interactions in one gene, but also in the SNP interactions in different genes, and has been reported in studies on cancer [149], obesity [150], and psoriasis [151]. Svinareva [152] studied the risk of developing POAG by SNP–SNP interactions and found that MMP-1 rs1799750 and MMP-9 rs2250889 were the most affected, especially in patients with POAG. There have been several studies on the relationship between SNP-to-SNP interactions and glaucoma in various genes, but those on MMP genes are still insufficient, so further studies are needed.

**Table 2 biomolecules-12-01368-t002:** Matrix metalloproteinase (MMP) gene single-nucleotide polymorphisms (SNP) likely to affect glaucoma subtypes.

Genes	SNP	Risky Glaucoma Type	Protective Glaucoma Type	Systemic Risks Affecting Glaucoma	Clinical Characteristics	References
MMP-1	rs1799750	POAG (Multiple ethnicities) XFG (Asian)	-	Hypertension	Earlier onset of POAG Likely to cause optic nerve damage Likely to increase AH outflow resistance Abnormal accumulation of matrix in TM of POAG patients	[91,122,123,124,125,126,127,128,130,153,154]
MMP-9	rs3918249	POAG (Asian) PACG (Caucasian)	POAG (Caucasian) PACG (Multiple ethnicities) XFG (Caucasian)	-	Likely to decrease axial length of eye in PACG Likely to increase IOP	[126,131,132,133,137,139]
	rs2250889	POAG (Caucasian) PACG (Asian) XFG (Caucasian)	-	-	Elevation of plasma MMP-9 level Decreased IOP in POAG	[126,132,137]
	rs17576	POAG (Asian) PACG (Asian, Caucasian)	POAG (Caucasian)	Hypertension Diabetes	Elevation of plasma MMP-9 level Decreased IOP in POAG	[122,126,131,133,137,139,140,141,155]
	rs3918254	PACG (Asian)	-	-	-	[135]
	rs2274755	NTG (Asian)	-	-	Likely to cause weakness in lamina cribrosa	[144]
	rs3918242	POAG (Asian) PACG (Asian)	POAG (Caucasian)	-	Elevation of plasma MMP-9 level	[124,126,145,146]

Primary open-angle glaucoma (POAG); primary angle closure glaucoma (PACG); normal tension glaucoma (NTG); pseudoexfoliation glaucoma (XFG); aqueous humor (AH); trabecular meshwork (TM).

### 2.3. Matrix Metalloproteinases and Wound Healing in Glaucoma Surgical Treatments

MMPs are expressed in various eye tissues and are closely associated with wound modulation [156]. The conjunctiva is the outermost tissue of the eye, and many studies have shown that MMPs have a great influence on conjunctival diseases, wound healing, and treatment in patients [111,157,158,159,160]. Glaucoma filtration surgery (GFS) is performed in patients who do not respond sufficiently to glaucoma medications or cannot be treated with drugs. The success rate of GFS is closely related to the condition of the conjunctiva [161,162]. The purpose of GFS is to control IOP by creating a reservoir for AH (filtering bleb) in the subconjunctival area by creating a new drainage route through the sclera [163,164]. The most common reason for GFS failure is scarring of the filtering bleb [165]. Surgery damages the structure and wound healing process of the conjunctiva, which is followed by the activation of cell proliferation [166], elevated expression of ECM proteins [42], and inflammatory changes [167,168]. Fibroblasts proliferate in response to various cytokines (TGF-β, IL-6, and IGF-1) and cell-mediated reactions by macrophages and neutrophils [169,170]. Finally, the wound-healing process leads to ECM remodeling, and MMPs cause the ECM to contract afterward, resulting in fibrosis and tissue scarring [171]. In some cases of GFS, excessive wound healing occurrs in the subconjunctival area (Tenon’s capsule). The over-proliferation of fibroblasts causes inflammatory changes in the ECM and ECM overproduction. MMPs and TIMPs are activated through these changes in the ECM environment, causing ECM contraction [172]. Excessive wound healing after GFS causes scarring and fibrosis in the newly formed drainage site and filtering blebs, which compromises IOP control and leads to GFS failure [173,174,175]. Currently, using mitomycin C (MMC) or 5-Fluorouracil during or after surgery is the gold standard for suppressing these fibrotic changes and increasing the surgical success rate, but these have the risk of causing other complications (hypotony, endophthalmitis, blebitis, bleb leakage, etc.) [176,177,178].

Several studies have been conducted to identify changes in MMPs after GFS. In a rabbit model of GFS, MMPs (MMP-1, -2, -3, and -9 and MT1-MMP) were detected in the conjunctiva a few days after surgery [42]; however, Liu et al. [175] also conducted a study with rabbits and reported that only the expression of MMP-1, -2, and -14 was detected, whereas MMP-3 and -9 expression were not detected. These conflicting results may be attributed to the different types of rabbits used in each study. However, a rat model study [171] showed that the expression of MMP-1 and -3 was increased after GFS, which is similar to the findings of Shima et al. In another study, MMP-1 and TIMP-1 were detected in human subconjunctival tissue that underwent GFS, but not in controls [179]. Similar to a previous study, two other studies [180,181] also confirmed the expression of MMP-1, but the results of these two studies were contradictory regarding the expression of other MMPs following GFS. McCluskey et al. [180] reported extensive expression of MMP-1, -2, and -3 in specimens, but Välimäki et al. [181] reported significantly reduced expression of MMP-1, -2, and -3, but TIMP-3 immunoreactivity was significant, while the other study only reported TIMP-2 activity. The conflicting results of the two studies may have been affected by the relatively small sample size (10 vs. 3), and the subtypes of glaucoma and the postoperative period of filtering blebs were not controlled in each study.

A prospective randomized clinical trial study of Korean patients who underwent GFS reported that increased MMP-9 levels in Tenon’s capsule collected during surgery were a risk factor for GFS failure [182]. Helin-Toiviainen et al. [183] reported similar results of increased MMP levels compared to the control; however, the GFS success rate was not related to the elevation of MMP levels. The different results of these studies may be due to the previous use of glaucoma medications, which increase MMP activity [157,183].

In a recent study, it was confirmed that the expression of MMP-2 changed when the depth of dissection of the subconjunctival tissue varied during GFS in a rabbit model. When dissecting the tissue, the subepithelial region of the conjunctiva, which is shallower than the episcleral region, showed decreased MMP-2 expression, which is associated with decreased fibrotic changes and angiogenesis [184]. Similar to the studies mentioned above, existing studies have shown that scarring or fibrotic changes in filtering blebs can be decreased as the amount of conjunctival manipulation is minimized during GFS [166,185,186].

Research has been published on several MMP inhibitors that control fibrotic changes after GFS. An in vitro study using broad-spectrum MMP inhibitors (Ilomastat, CellTech, BB-94), which inhibit MMP-1, -2, and -3, significantly reduced ECM production and contraction in human Tenon’s capsule fibroblasts without cell toxicity [187]. Other studies using ilomastat as an MMP inhibitor in animal models with GFS reported prevention of ECM contraction and similar bleb survival rates as those using MMC, and ilomastat did not induce histological toxicity [188,189,190]. In another study, the sequential injection of saratin (a polypeptide with antithrombotic and anti-inflammatory properties), bevacizumab (monoclonal anti-VEGF), and ilomastat was performed in the functioning blebs of rabbits with GFS. Functioning bleb survival was prolonged by injections, similar to that observed in the MMC injection group [191]. Doxycycline acts as an inhibitor of MMP activity [192,193,194] and has been investigated in a rabbit GFS model [195]. However, it only showed some extent of MMP-1 and -9 inhibition, which was not definite, and further studies are needed.

In addition to agents that directly inhibit MMPs, research results describing agents that indirectly inhibit MMPs have also been published. Cinik et al. [196] conducted an experiment using everolimus during GFS in a rabbit model and compared the results with those of MMC and control groups. Everolimus has immunosuppressive and antiproliferative properties and is currently used to prevent rejection in organ transplants and as a targeted therapy for many types of cancers [197,198,199,200]. The expression of MMP-2 and -9 was significantly lower in the everolimus group than in the other groups, and the mean IOP level remained lower than that in the other groups for four weeks [196]. 

Wang et al. [201] conducted an in vitro study using olmesartan in human Tenon’s capsule fibroblasts and reported reduced expression of MMP-2 mRNA, but increased expression of TIMP-1 and -2 mRNAs in a dose-dependent manner. Olmesartan acts as an angiotensin II receptor blocker and is currently used to treat hypertension [202]. In addition to altered mRNA expression, olmesartan decreased cellular neovascularization and proliferation [201]. 

Retinoic acids inhibit TGF-β activity, resulting in anti-inflammatory and antifibrotic properties [203]. The action of retinoic acid is mediated by retinoic acid receptors [204]. Liu et al. [205] reported MMP-1 and -3 expression decreased after induction with a retinoic acid receptor agonist in human Tenon’s capsule fibroblasts, and reduced IOP levels were observed in GFS-treated rats. 

Secreted protein acidic and rich in cysteine (SPARC) regulates ECM–cell interactions, without contributing to ECM structures [206]. SPARC regulates MMP expression and significantly influences ECM remodeling through its interaction with growth factors and ECM proteins [207]. In a GFS rat model, the SPARC knockout group showed an improved surgical success rate compared with the control group [208]. This result was supported by an in vitro study of SPARC knockout in human Tenon’s capsule fibroblasts, which was performed by the same researchers. Expression of MMP-2, -9, -14, and several inflammatory cytokines was reduced in the SPARC knockout group [207,209,210,211] (Figure 5).

### 2.4. Ocular Surface Change by Matrix Metalloproteinase Function during Medical Treatment for Glaucoma

The initial treatment for glaucoma is medication; in most cases, eye drops are frequently used [213]. However, it is also true that many patients experience discomfort when using glaucoma drugs, and their adherence to treatment is poor [214]. Complex drug use methods, economic problems, and lack of knowledge about the disease also affect drug adherence; however, the discomfort felt when using drugs also has a major impact [215,216,217]. Preservatives in topical glaucoma medications, such as polyquaternium-1 (PQ-1) and benzalkonium chloride (BAK), may disrupt the ocular surface, activate proinflammatory cytokines, cause squamous metaplasia of the conjunctival epithelium, fibrosis of Tenon’s capsule, and decrease the number of goblet cells [217,218]. The long-term use of topical medications may induce subclinical or clinical conjunctival inflammatory changes and reduce the success rate of GFS [219]. The cornea is a major part of the ocular surface and is a tissue that undergoes minute remodeling during homeostasis. Corneal injury upregulates various proteins, including MMPs, which reform the collagen lattice [220]. In particular, MMP-1, -2, -3, and -9 repair the corneal epithelium and stroma [221]. An imbalance in MMP and TIMP activity in the cornea is likely to cause a keratolytic process and disrupt the corneal basement structure, leading to corneal ulcerations [222]. Patients on anti-glaucoma medications, especially topical prostaglandins and herpetic simplex keratitis (HSK), are one of the most severe ocular surface complications that can cause corneal ulceration or ‘corneal melt’ [223,224].

Several studies have reported that inflammatory changes are associated with the use of preservative-containing topical glaucoma medication. A rabbit model using preservative-containing glaucoma medication showed increased expression of MMPs (MMP-1 and -9) and decreased activity of TIMP-1 [225]. Patients using preservative-containing glaucoma medications also show increased MMP-1 and MMP-9 saturation in tears [226]. However, patients using preservative-free glaucoma medications show relatively decreased levels of MMP-9 on the ocular surface [43]. A study of Korean glaucoma patients showed similar results of increased MMP-9 activity in tear films when preservative-containing topical drugs were used [227]. Interestingly, studies using animal models have shown that glaucoma medication itself may affect MMP activity and cause changes in the ocular surface and conjunctival tissues. Prostaglandin analog medication without preservatives caused elevated MMP-3 and -9 expression and induced inflammatory change in the ocular surface of mice similar to dry eye [228]. Other studies using rat models have reported that prostaglandins, α-blockers, β-blockers, and α2-agonists promote the modulation of MMP and TIMP activity, which degrades the ECM in ocular surface tissue [229]. 

## 3. Conclusions

In conclusion, MMPs are involved in the pathophysiological mechanisms and treatment outcomes of glaucoma. MMP activity in ocular tissues plays an important role in the regulation of AH outflow, which affects the IOP. MMP activity can disrupt ECM in the retina, leading to glaucomatous optic nerve head changes. Various MMP genetic pleomorphisms are associated with glaucoma, and MMP genetic traits may play a role in predicting the onset or progression of glaucoma by classifying the risk of developing glaucoma in variants of the MMP gene. MMPs affect both surgical outcomes and compliance with medical treatment in patients with glaucoma.

Therefore, further studies on glaucoma and MMPs should be conducted on various topics, including etiology and treatment. In particular, research on MMP, single SNP, and glaucoma has been conducted in many studies, but future studies are still needed, and more attention is needed on the gene–gene interaction of MMPs and glaucoma. Future studies will likely reveal the function of MMPs in glaucoma and will be able to play a role in preventing glaucoma and the successful treatment of glaucoma. 

## Figures and Tables

**Figure 1 biomolecules-12-01368-f001:**
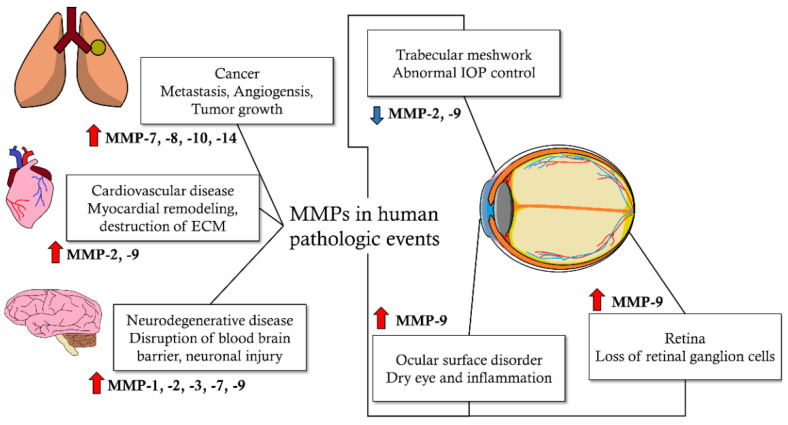
Effect of matrix metalloproteinases (MMPs) on human pathologic events [12,13,14,15,16,17]. In cancer, metastasis, angiogenesis, and tumor growth are affected by upregulated MMP-7, -8, -10, and -14 activity. Myocardial remodeling and extracellular matrix (ECM) destructions occur in cardiovascular disease by increased MMP-2 and -9 activity. Neuronal injury and disruption of the blood–brain barrier is caused by elevated MMP-1, -2, -3, -7 and -9 activity in neurodegenerative disease. Additionally, in the eye, increased MMP-9 activity affects ocular surface disorder, which can cause dry eye and surface inflammatory change. Furthermore, increased MMP-9 activity can cause retinal ganglion cell death in the retina. Decreased MMP-2 and -9 activity can disrupt homeostasis of intraocular pressure (IOP) in trabecular meshwork.

**Figure 2 biomolecules-12-01368-f002:**
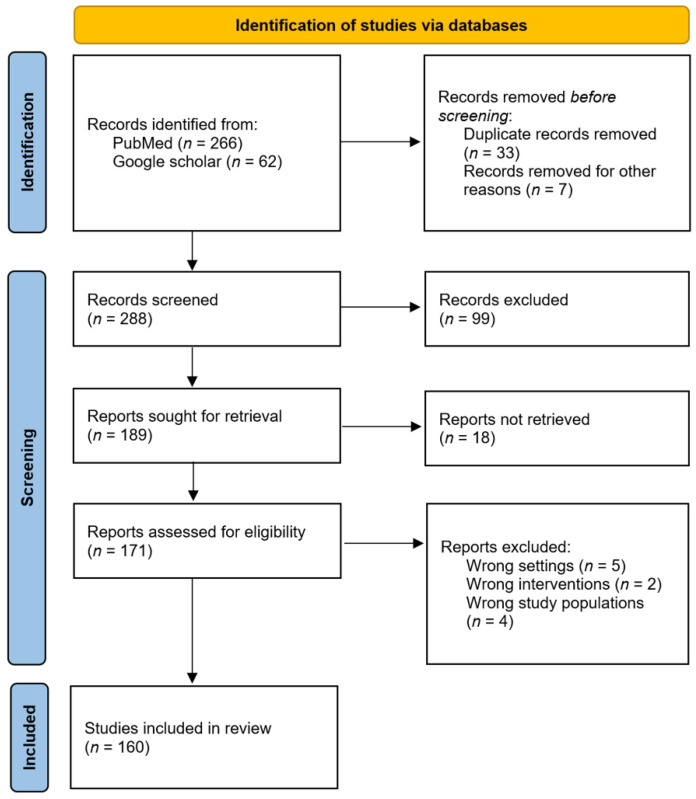
Selection of eligible studies.

**Figure 3 biomolecules-12-01368-f003:**
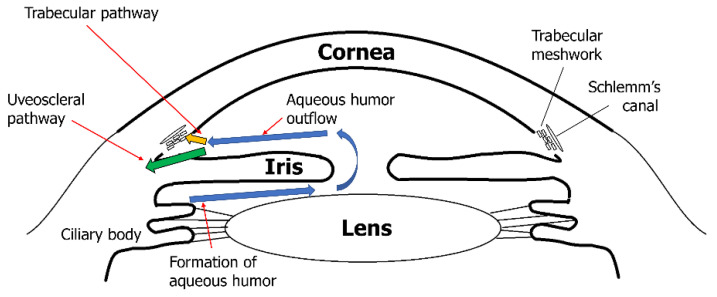
Aqueous humor production and outflow in the eye. Aqueous humor (AH) is produced in the ciliary body. After AH enters the posterior chamber of the eye, it flows through the pupil and enters the anterior chamber of the eye (blue arrows). About 60 to 80% of AH outflows to the trabecular pathway (yellow arrow) and others pass through the uveoscleral pathway (green arrow). Increased matrix metalloproteinase (MMP) activity in the trabecular meshwork elevates AH outflow, which in turn reduces intraocular pressure (IOP). However, decreased MMP activity in the trabecular meshwork decreases AH outflow, which causes elevated IOP [63,65,71,72,73].

**Figure 4 biomolecules-12-01368-f004:**
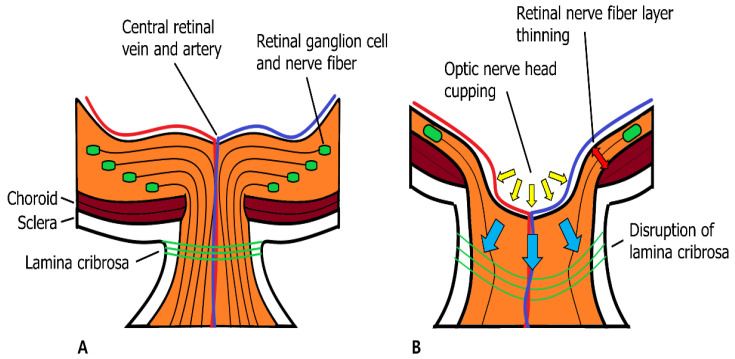
Normal retina and optic nerve head (ONH) (**A**). Retinal ganglion cell (RGC) axon bundle passes lamina cribrosa (LC) and exits the eye. LC supports the RGC axon bundles as they become the optic nerve. Glaucomatous ONH change (**B**). Increased intraocular pressure causes mechanical stress in the LC. Due to mechanical stress in LC, cytokine activity is increased, which induces matrix metalloproteinase (MMP) expression and extracellular matrix (ECM) remodeling in the ONH. Additionally, mechanical stress cause apoptosis of RGC by increased MMP-9 activity. The aftermath of glaucomatous ONH changes includes cupping of ONH (yellow arrows), retinal nerve fiber layer thinning by RGC apoptosis (red double arrow), and lamina cribrosa disruption (blue arrows) [31,95,96,97,98].

**Figure 5 biomolecules-12-01368-f005:**
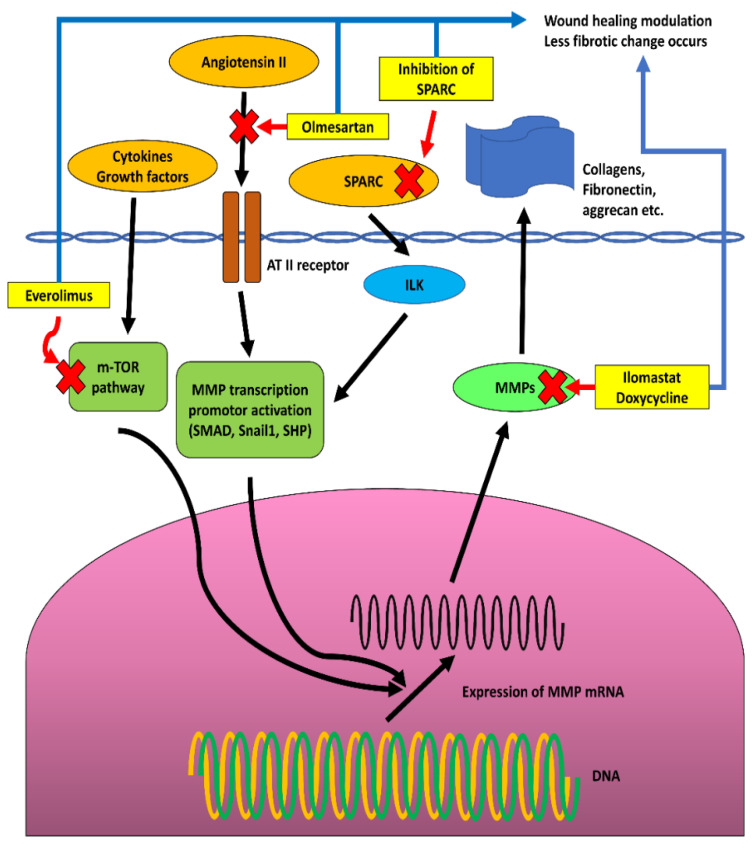
Recent studies about drugs that inhibit matrix metalloproteinase (MMP) activity shown in summarized figures [188,194,196,201,205,209,210,211]. Integrin linked kinase (ILK), Small heterodimer partner (SHP), Mechanistic target of rapamycin (m-TOR), Suppressor of mothers against decapentaplegic (SMAD), Secreted protein acidic and rich in cysteine (SPARC), Angiotensin II receptor (AT II receptor), Snail1 is a member of the Snail superfamily of zinc-finger transcription factors [212]. Inhibiting agents (yellow box) and indication of inhibiting pathways (red cross and red arrow).

**Table 1 biomolecules-12-01368-t001:** Matrix metalloproteinase (MMP) subtypes and substrates.

MMP Subtypes	MMP Number	Substrates
Collagenases [20,21,22]	MMP-1	type I, II, III, VI, and X collagens; aggrecan; and entactin
	MMP-8	type I, II, and III collagens and aggrecan
	MMP-13	type I, II, and III collagens
Gelatinases [20,23,24]	MMP-2	type I, IV, V, VII, X, and XI collagens; vitronectin; fibronectin; aggrecan; laminin; elastin; and tenascin C
	MMP-9	type IV, V, and XIV collagens; entactin; vitronectin; aggrecan; and elastin
Stormelysins [20,25,26]	MMP-3	type III, IV, IX, and X collagens; aggrecan; fibronectin; laminin; tenascin C; and vitronectin
	MMP-10	type IV collagen; aggrecan; and fibronectin
	MMP-11	type IV collagen; fibronectin; laminin; and aggrecan
Membrane-type MMPs [20,27]	MMP-14	Activator of pro-MMP-2; type I, II, and III collagens; vitronectin; fibronectin; and laminin-1
	MMP-15	Activator of pro-MMP-2; tenascin; fibronectin; and aggrecan
	MMP-16	Activator of pro-MMP-2; fibronectin; and type III collagen
	MMP-17	Fibrin and cleaves pro-TNF-α
	MMP-24	Activator of pro-MMP-2
	MMP-25	type IV collagen; gelatin; fibronectin; and fibrin
Metalloelastases [20,26,28]	MMP-12	type IV collagen; elastin; proteoglycan; fibronectin; laminin; and vitronectin
	MMP-19	cartilage oligomeric matrix protein and aggrecan
	MMP-20	enamel matrix proteins
	MMP-28	cartilage oligomeric matrix protein and aggrecan
Other MMPs [20,26,29]	MMP-7	aggrecan; fibronectin; laminin; and type IV collagen
	MMP-26	fibronectin; type IV collagen; fibrinogen; and gelatin

## Data Availability

No datasets were generated or analyzed during the current study.

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
