# Peer review of "Matrix Metalloproteinases and Glaucoma"

_biomolecules, 2022, doi:10.3390/biom12101368_

Round 1

Reviewer 1 Report

the review is devoted to an important and urgent problem - the analysis of the relationship of MMP with glaucoma. The review contains a lot of factual material on the problem under study. However, the section devoted to the genetic part (3.3. Genetic polymorphism in matrix metalloproteinase and glaucoma subtypes) needs to be substantially revised:

1.    should beexclude from this section the material on the pathogenesis of glaucoma (it must be moved to the appropriate section).

 2. it is necessary to significantly expand the references list used to describe this section - to the 14 sources currently available on the theme of "SNP MMP and glaucoma", it is necessary to add another 10 following missing sources ():

1)Chen M, Yu X, Xu J, Ma J, Chen X, Chen B, Gu Y, Wang K. Association of Gene Polymorphisms With Primary Open Angle Glaucoma: A Systematic Review and Meta-Analysis. Invest Ophthalmol Vis Sci. 2019; 1;60(4):1105-1121. doi: 10.1167/iovs.18-25922.

2)Majsterek I, Markiewicz L, Przybylowska K. Association of MMP1-1607 1G/2G and TIMP1 372 T/C gene polymorphisms with risk of primary open angle glaucoma in a Polish population. Med Sci Monit. 2011;17(7):CR417–CR421. doi:10.12659/msm.881854.

3)Micheal S, Yousaf S, Khan MI, et al. Polymorphisms in matrix metalloproteinases MMP1 and MMP9 are associated with primary open-angle and angle closure glaucoma in a Pakistani population. Mol Vis. 2013; 19: 441–447. PMID: 23441116

4)Zhao, F., Fan, Z., & Huang, X. (2020). Role of matrix metalloproteinase-9 gene polymorphisms in glaucoma: A hospital-based study in Chinese patients. Journal of clinical laboratory analysis34(3), e23105. https://doi.org/10.1002/jcla.23105

5)Zhang Y, Wang M, Zhang S. Association of MMP-9 Gene Polymorphisms with Glaucoma: A Meta-Analysis. Ophthalmic Res. 2016;55(4):172-9. doi: 10.1159/000443627.

6)Tsironi EE, Pefkianaki M, Tsezou A, et al. Evaluation of MMP1 and MMP3 gene polymorphisms in exfoliation syndrome and exfoliation glaucoma. Mol Vis. 2009;15:2890‐2895. Published 2009 Dec 25.

7)Awadalla MS, Burdon KP, Kuot A, Hewitt AW, Craig JE. Matrix metalloproteinase-9 genetic variation and primary angle closure glaucoma in a Caucasian population. Mol Vis. 2011;17:1420‐1424.

8)Svinareva DI. The contribution of gene-gene interactions of polymorphic loci of matrix metalloproteinases to susceptibility to primary open-angle glaucoma in men. Research Results in Biomedicine. 2020;6(1):63-77. (In Russian) DOI: 10.18413/2658-6533-2020-6-1-0-6

9)Saleh, V.M., Auda, I.G., Ali, E.N. The functional polymorphism -863 C/A in the TNF-α gene is associated with primary open-angle glaucoma development in Iraqi patients (2022) Gene Reports, 28,  № 101653,

10)Ponomarenko I, Reshetnikov E, Dvornyk V, Churnosov M. Functionally significant polymorphisms of the MMP9 gene are associated with primary open-angle glaucoma in the population of Russia. Eur J Ophthalmol. 2022 Mar 7:11206721221083722. doi: 10.1177/11206721221083722.

3.Along with the main effects of MMP genes SNP in the formation of glaucoma, it is necessary to consider the role of intergenic interactions of MMP genes SNPs, since different variants of MMPs are in contact with each other and this may be based on genetic interactions

4.Data should also be provided on the impact of MMP loci on several clinical characters of the disease: intraocular pressure, age of the disease manifestation, etc.

5. it is necessary to discuss not only the direct effect of MMP polymorphisms on the pathogenesis of glaucoma, but also their indirect effect on the risk of developing the disease through various risk factors (for example, MMP loci have a risk value for arterial hypertension, which in turn is a risk factor for glaucoma)

6. It is necessary to consider the relationship of MMP polymorphisms with glaucoma through the functional effects of the studied polymorphisms and bioinformatic data on the relationship of MMP with the biological pathways of glaucoma

7. it is necessary to exclude errors and inaccuracies in the work (for example, the link to source 104 is duplicated, position 282)

8. may be the results of this section should be presented in the form of a table (summary data on associations of MMP polymorphisms with various subtypes of glaucoma)

Reviewer 2 Report

The review article by Kim MH et al entitled “Matrix Metalloproteinases and Glaucoma” discusses, in general, the relationship between Matrix Metalloproteinases (MMP) and Glaucoma which further details genetic pleomorphism, pathogenesis, wound healing in surgical treatments, and ocular surface changes during medical treatment. The topic discussed here is interesting given the fact that MMPs are ubiquitous in the body and have essential roles in both normal physiology and pathology. In the eye, both MMPs and the MMP/TIMP ratio in ocular tissues involved in aqueous humor outflow have an important role in the regulation of intraocular pressure (IOP). The article is well written but needs a few amendments as follows.

1)      The authors may revisit the Material and Methods Section and make it concise or simply make it part of the introduction. Given this is a review article, the Material and Methods section and Figure2 can be reduced.

2)      Section 3.1 can also be merged with the introduction part.

3)      Section 3.2.1. This section mainly describes the Matrix metalloproteinase subtypes and the substrate they act on which is collated in Table 1. This can be added to the Introduction section since it describes MMPs in general and not MMPs concerning the pathogenesis of Glaucoma precisely.

4)      Section 3.2.2 and 3.2.3. It would be more informative and engaging for readers if the authors provide a graphical representation of the section.  

5)      Page 11 Figure2 which is Figure 3. Also, it is highly recommended to the authors rewrite the legends for all Figures and Tables provided in the manuscript.

6)      Although the manuscript covers MMPs in glaucoma-associated ocular tissue but there are reports of activation of MMPs in nontarget ocular tissues such as the cornea. Because the stroma of the cornea is largely composed of collagen, MMP activation in the cornea might be expected to cause stromal ECM remodeling and corneal thinning. MMP activity in the conjunctiva might help prevent treatment-related changes in the conjunctival ECM. If authors can elaborate on this topic, it might add more weight and information to the article.

7)      The conclusion section is vaguely described (line 477, we confirmed the MMP and Glaucoma are closely associated) this is not an original research article. Highly advised to revisit this section and make changes and most importantly describes future directions which is part of ant review article.

8)      Moderate English language correction is required for Typos and grammatical mistakes in the manuscript.

Round 2

Reviewer 1 Report

The authors have made all the necessary changes in the article.